# TRIB3 Mediates Fibroblast Activation and Fibrosis though Interaction with ATF4 in IPF

**DOI:** 10.3390/ijms232415705

**Published:** 2022-12-11

**Authors:** Lan Wang, Wenyu Zhao, Cong Xia, Zhongzheng Li, Weiming Zhao, Kai Xu, Ningdan Wang, Hui Lian, Ivan O. Rosas, Guoying Yu

**Affiliations:** 1State Key Laboratory Cell Differentiation and Regulation, Henan International Joint Laboratory of Pulmonary Fibrosis, Henan Center for Outstanding Overseas Scientists of Pulmonary Fibrosis, College of Life Science, Henan Normal University, Xinxiang 453007, China; 2Division of Pulmonary, Critical Care and Sleep Medicine, Baylor College of Medicine, Houston, TX 77030, USA

**Keywords:** idiopathic pulmonary fibrosis, *TRIB3*, *ATF4*, epithelial cell, fibroblast activation

## Abstract

Idiopathic pulmonary fibrosis (IPF) is a fatal interstitial lung disease characterized by fibroblast activation, excessive deposition of extracellular matrix, and progressive scarring; the pathogenesis remains elusive. The present study explored the role of Tribbles pseudokinase 3 (*TRIB3*), a well-known stress and metabolic sensor, in IPF. *TRIB3* is down-regulated in the lungs of IPF patients in comparison to control subjects. Deficiency of *TRIB3* markedly inhibited A549 epithelial cells’ proliferation and migration, significantly reducing wound healing. Conversely, overexpression of *TRIB3* promoted A549 cell proliferation and transmigration while it inhibited its apoptosis. Meanwhile, overexpressed *TRIB3* inhibited fibroblast activation and decreased ECM synthesis and deposition in MRC5 cells. *TRIB3* attenuated pulmonary fibrosis by negative regulation of ATF4, while *TRIB3* expression markedly inhibited *ATF4* promoter-driven transcription activity and down-regulated *ATF4* expression. A co-culture system showed that *TRIB3* is important to maintain the normal epithelial–mesenchymal crosstalk and regulate fibroblast activation. Taken together, our data suggested that an axis of *TRIB3–ATF4* is a key mediator in IPF which might be a potential target for fibroproliferative lung disease treatment.

## 1. Introduction

Idiopathic pulmonary fibrosis (IPF), one of the most common forms of interstitial lung disease (ILD), is a chronic, progressive, and usually lethal lung disease of unknown etiology and refractory to current therapeutic options [1]. The main pathological features of IPF include epithelial injury [2], the recruitment of inflammatory cells [3,4], the aberrant differentiation and proliferation of fibroblasts [5], and the persistence of apoptosis-resistant myofibroblasts [6] in fibrotic lesions. Resident lung fibroblasts-derived myofibroblasts are the major contributors to the processes of ECM deposition and tissue distortion in IPF. Stimulation with fibrotic factors, such as including transforming growth factor beta (*TGF-β*), platelet-derived growth factor (*PDGF*), and connective tissue growth factor (*CTGF*), promote fibroblast recruitment and myofibroblast activation, which are characterized by a spindle or stellate morphology with α-smooth muscle actin (*α-SMA*) stress fibers coupled with a hypersecretion phenotype, due to which they produce copious amounts of fibrillary extracellular matrix (ECM) proteins, such as collagen and fibronectin. Dysregulated crosstalk between the epithelium and the mesenchymal cells regulates lung development and homeostatic equilibrium [7] and further forms a profibrotic milieu and hampers the normal alveolar wound-repair process [8,9,10,11]. Until now, the molecular mechanisms by which alveolar epithelial cells (AECs) become activated and communicate with the fibroblast lead to fibrosis are not fully understood.

Tribbles pseudokinase 3 (*TRIB3*, also known as *TRB3*, *NIPK*, and *SKIP3*) is the mammalian homolog of Drosophila Tribbles [12]. TRIB3 is a pseudokinase, which contains a Ser/Thr protein kinase-like domain, but lacks the ATP-binding pocket and catalytic residues [13]. TRIB3 can serve as a molecular scaffold for the assembly of coactivator and corepressor complexes [14] and has been demonstrated to interact with several transcriptional mediators such as CCAAT-enhancer-binding protein homologous protein (*CHOP*), peroxisome proliferator-activated receptor alpha (*PPARα*), and activating transcription factor 4 (*ATF4*) [15,16]. Through those interactions, TRIB3 coordinates crucial cellular processes such as glucose and lipid metabolism, adipocyte differentiation, or apoptosis [15,16]. Alterations in *TRIB3* gene expression have also recently been linked to numerous chronic diseases, including diabetes, hepatitis, and tissue fibrosis [17,18,19,20]. Previous study demonstrated that TRIB3 interacting with ATF4 inhibits its transcription activity [21,22,23], participating in regulating insulin exocytosis in human and mouse beta cells [21] and Parkinson’s disease process [24]. Elevated *TRIB3* expression was observed in alveolar macrophages (AMs) from bleomycin challenged fibrotic mice, which contributes to pulmonary fibrosis (PF) [25].

Herein, we found that *TRIB3* expression was down-regulated in the lungs of IPF and negatively associated with profibrotic gene expression. TRIB3 can enhance epithelial cell migration and inhibit lung fibroblast activation in vitro. The expression of TRIB3 was negatively correlated with disease severity in IPF patients. TRIB3 expression negatively regulated ATF4 expression, which promoted fibroblast proliferation, migration, and activation contributing to pulmonary fibrosis. These findings suggested that the TRIB3–ATF4 axis is a key mediator in IPF, which may provide novel insights into the pathogenesis of IPF.

## 2. Results

### 2.1. TRIB3 Expression Was Downregulated in the Lung Tissue of Patients with IPF

To verify expression of *TRIB3*, we initially evaluated the expression of *TRIB3* in healthy and IPF lungs to properly describe human disease-relevant alterations from the Gene Expression Omnibus (GEO) RNA-seq dataset (GSE32537) and found the *TRIB3* expression was significantly down-regulated in IPF patients in comparison to normal donors (Figure 1A). *TRIB3* expression was negatively correlated with the expressions of *COL1A1*, *COL1A2*, *FN1*, and *ACTA2* in the RNA-seq dataset (GSE32537) by Spearson correlation coefficient (Figure 1B). This significant decrease in the *TRIB3* transcript level was validated by the qPCR (Figure 1C). Consistently, western blotting showed that *TRIB3* protein levels in the lung tissue of patients with IPF dramatically decreased in comparison to normal lung tissues (Figure 1D). These findings exhibited that *TRIB3* was stably repressed in IPF and negatively associated with pulmonary fibrosis markers.

### 2.2. TRIB3 Was Essential for Wound Healing in Epithelial Cells 

To determine the role of *TRIB3* in epithelial cells, we examined the impact of *TRIB3* deficiency on the proliferation and migration of A549 cells. Knockdown of *TRIB3* by shRNA in the epithelial cells significantly inhibited A549 cells’ proliferation (Figure 2A,B). Moreover, *TRIB3* knockdown significantly reduced wound healing after 48 h, indicating TRIB3 suppressed the epithelial cell migration (Figure 2C). In addition, *TRIB3* knockdown inhibited profibrotic gene transcription (Figure 2D). Silencing of TRIB3 reduced the protein level of α-SMA, while it enhanced the E-Cadherin expression (Figure 2E). Conversely, overexpression of *TRIB3* in A549 cells increased the proliferation and transmigration of A549 cells, while it inhibited apoptosis of A549 cells (Figure 3A–E), and *TRIB3* overexpression promoted *ACTA2* and *VIM* transcriptions and protein levels of *α-SMA*, *VIM*, and *p-SMAD3*, as well as inhibiting the expression of E-Cadherin (Figure 3F,G). These observations, taken together, suggested that *TRIB3* is crucial in modulating epithelial cell proliferation, migration, and apoptosis.

### 2.3. TRIB3 Overexpression Inhibited the Fibroblast Activation 

To further understand more about how *TRIB3* could be involved in the development of pulmonary fibrosis in vitro, we overexpressed *TRIB3* in MRC5 cells for 48 h and demonstrated that TRIB3 significantly decreased the profibrotic genes expression, including *ACTA2*, *COL1A1*, and *FN1* at the mRNA level and the protein level of collagen I, fibronectin, and α-SMA in MRC5 cells (Figure 4A,B). The fibroblast-induced collagen gel contraction (CGC) assay demonstrated that TRIB3 overexpression inhibited the fibroblast contractility (Figure 4C) and the expression of vimentin, a marker for epithelial–mesenchymal transition (EMT) (Figure 4D). Taken together, our findings showed that increased TRIB3 expression may assist in reducing lung fibrosis in vitro.

### 2.4. TRIB3 Interaction with ATF4 Mediated Lung Fibroblast Activation

To investigate the molecular mechanisms of *TRIB3* attenuating lung fibrosis, we postulated that *TRIB3* plays a momentous role in IPF through regulation of *ATF4*. As shown in Figure 5A, immunoprecipitation and silver staining assay demonstrated that TRIB3 directly interacts with *ATF4* in MRC5 cells. To further demonstrate this relationship, we examined the above associations using co-immunoprecipitation and western blotting and found that *TRIB3* and *ATF4* co-immunoprecipitated (Figure 5B). We also found that *TRIB3* overexpression inhibited *ATF4* expression at RNA and protein levels in MRC5 cells (Figure 5C,D). In addition, to further verify whether *TRIB3* affects the transcription of *ATF4*, we established two luciferase-reporter vectors, including *ATF4* transcription start sites (TSSs) (−1000 bp to +1000 bp), as described in Methods. Overexpression of *TRIB3* markedly inhibited *ATF4* promoter-driven luciferase activity (Figure 5E). We next tested whether *ATF4* overexpression influences fibroblast activation and found that overexpression of *ATF4* promoted *ACTA2*, *COL1A1*, and *FN1* transcriptional activity (Figure 6A). Western blotting confirmed that *ATF4* overexpression promotes ECM deposition and increases the proliferation and contraction of fibroblasts (Figure 6B,C), details as described in Methods. Furthermore, immunofluorescence staining showed that ATF4 overexpression increased the vimentin expression (Figure 6D). Altogether, these findings indicate that the enhanced expression of TRIB3 influences lung fibroblast activation by regulation of *ATF4* expression.

### 2.5. TRIB3 Maintained Epithelial-Mesenchymal Crosstalk and Attenuated Fibrosis by Negatively Regulation of ATF4 

To evaluate whether alteration of *TRIB3* expression in epithelial cells regulates the fibroblast activation through down-regulation of *ATF4* expression, a co-culture system was established and the conditioned media (CM) were collected from A549 cells, for which *TRIB3* was overexpressed, for fibroblast MRC5 cells culture (Figure 7A). Interestingly, this CM not only suppressed the *ATF4* expression but also inhibited the profibrotic gene expressions in lung fibroblasts at both RNA and protein levels (Figure 7B,C). Putting it all together, these findings indicate that *TRIB3* plays a pivotal role in maintaining normal epithelial–mesenchymal crosstalk and negatively attenuating fibrosis through interaction with *ATF4* (Figure 8).

## 3. Discussion

Increasing studies suggest that AECs injury predominately contributes to the development and progression of IPF. Dysfunction of AECs is considered to be involved in injury-impairment, an early event and cascade to fibroblast response in IPF. In the present study, we demonstrated that TRIB3 contributes to the A549 epithelial cells’ proliferation and migration and repair; TRIB3 attenuates pulmonary fibrosis by interacting with ATF4. TRIB3 is associated with the epithelial–mesenchymal crosstalk and regulates fibroblast activation. We provide a potential therapy strategy for fibroproliferative lung disease.

Little is known about the regulation of TRIB3 expression in IPF. A growing body of evidence suggests that TRIB3 is involved in the regulation of various biological processes that are relevant to cancers [26,27,28,29]. For instance, elevated TRIB3 links stress signals to induce breast cancer initiation and progression by supporting breast cancer stemness, and elevated TRIB3 is positively associated with epidermal growth factor receptor (EGFR) stability and non-small cell lung cancer (NSCLC) progression [26,29]. To wonder whether TRIB3 regulates IPF development through regulating EMT process, we altered TRIB3 expression in AEC2s and tested the EMT markers. In this work, we demonstrated that the down-regulated TRIB3 suppressed wound healing in epithelial cells, while up-regulation of TRIB3 promoted epithelial cell proliferation and migration. Meanwhile, blockage of TRIB3 inhibited the expression of α-SMA, conversely, TRIB3 overexpression increased the expression of α-SMA, vimentin, and p-Smad3 in epithelial cells. Hence, consistent with our initial hypothesis that TRIB3 may play an important role in epithelial cell migration by involvement of EMT and the down-regulation of TRIB3 in IPF, it may be detrimental to the repair of epithelial cells after injury. Furthermore, overexpressed TRIB3 in lung fibroblasts reduced α-SMA, vimentin, collagen I, and fibronectin expression, which are fibrotic markers and decreased ECM deposition. The co-culture system showed that overexpressed TRIB3 evidently inhibited fibroblast activation and ECM deposition, which is similar to overexpression of TRIB3 in fibroblasts. Previous study demonstrated that TRIB3 overexpression in AMs increased fibroblast activation [25]; our findings showed that TRIB3 may play different roles in various cells.

ATF4 is a basic leucine zipper transcription factor best known to be induced by stress responses. ATF4 has long been considered a profibrotic molecule because it enhances fibroblast proliferation, collagen synthesis, migration, and differentiation to myofibroblasts and induces fibroblasts’ apoptosis in tissue fibrosis [30,31,32,33]. Despite being less studied in lung disease, ATF4 was a profibrotic role in lung fibroblast activation and renal tubulointerstitial fibrosis [29,30]. For instance, ATF4 is an important regulator of metabolic reprogramming in myofibroblasts [29]. In fact, TRIB3 is a target of CHOP/ATF4; when up-regulating ATF4 expression, the level of TRIB3 was markedly increased [16,34]. Several studies have shown that TRIB3 interacts with ATF4 and inhibits ATF4 transcription activity [21,22,23]. To further explore the mechanism of TRIB3 attenuating fibroblast activation. We also demonstrated that TRIB3 interacts with ATF4 and inhibits ATF4 expression both at RNA and protein levels in a lung fibroblast and co-culture system through a negative feedback mechanism. Putting together, expression of TRIB3 by the epithelium is essential to recovery of the injured epithelium and suppression of mesenchymal cell activation by interaction with ATF4 and regulating its expression through a negatively regulated loop.

Even with our observations that TRIB3 regulation occurs in epithelium and fibroblast, we do acknowledge that our study has some limitations: whether TRIB3 mediated ATF4 signaling regulates EMT and fibrosis in IPF remains to be established in vivo. The possibility that TRIB3 functions through another signaling pathway in addition to the ATF4 signal in IPF has not been established yet. The underlying mechanism of TRIB3 in TGF-β-induced EMT is not fully unveiled and deserves further investigation. The role of TRIB3 in lung fibroblasts derived from IPF and its contribution to the progression of IPF requires an in-depth exploration in the future.

In conclusion, the key finding of our study is that TRIB3 promoted epithelial cell proliferation, migration, EMT, and attenuated lung fibroblasts’ abnormal activation. TRIB3 inhibited lung fibroblast activation by regulating ATF4 expression. Put together, these findings provide novel insights into the pathogenesis of the IPF and TRIB3–ATF4 axis, which might be an antifibrotic target for IPF management.

## 4. Material and Methods

### 4.1. Lung Tissue Sampling

Lung tissue samples for RT-qPCR and western blot analysis were obtained from the Henan Provincial Chest Hospital. Those included 23 lungs from patients with IPF and 15 normal lung histology samples from control (CTRL) subjects. The IPF samples were surgical remnants of biopsies or lungs explanted from patients with IPF undergoing pulmonary transplant. Controls were normal histology tissues obtained from normal disease-free margins of lung cancer resection specimens. IPF was diagnosed based on ATS/ERS/JRS/ALAT Clinical Practice Guidelines [35]. The clinical characteristics of all patients are summarized in Appendix A. All studies were approved by the Henan Provincial Chest Hospital Medical Research Ethics Committee (No. 2020-03-06). The research conformed to the principles of the Declaration of Helsinki. Oral and written informed consent was obtained from all patients. 

### 4.2. Cell Culture

A549 cells, MRC5 cells, and HEK293T cells were cultured in DME/F-12 and DMEM at 37 °C and 5% CO_2_. The medium was supplemented with 10% fetal bovine serum (FBS) 100 units/mL penicillin, 100μg/mL streptomycin, and 1 mM sodium pyruvate. 

### 4.3. Plasmid Construction, Lentivirus Package and Stable-Infected Cell Lines Construction

Flag-tagged TRIB3 and HA-tagged ATF4 were cloned into pcDNA3.1 vector by a standard subcloning procedure. Silencing TRIB3 was achieved by targeting the sequences 5′ GATCTCAAGCTGTGTCGCTTT 3′ (shTRIB3) in the pLKO.1 vector. Concentrate lentivirus particles were used to infect sub-confluent cultures in the presence of 5 μg/mL polybrene overnight. Twenty-four hours post-transfection, cells were selected in media containing 2 μg/mL puromycin. Details for primers are provided in Table 1.

### 4.4. Analysis of Cell Survival and Apoptosis 

Cell proliferation was measured by the CCK8 assay. Cells were plated into 96-well plates, and cell viability was detected by the CCK8 reagent purchased from the APExBLO company. Cells at a density of 2000 cells/well were plated in 6-well plates. After culturing for 14 days, the generated colonies were fixed with methanol for 30 min, stained with a hematoxylin solution, and photographed using a microscope. An Apollo567 in vitro Imaging Kit was purchased from RiboBio Corporation (Guangzhou, China) and was applied for the EdU incorporation assay. The apoptosis kit for TRIB3-overexpressed cells’ apoptosis detection is from Solarbio (CA1020). Annexin V and propidium iodide (PI) were used for labeling to assess cell apoptosis.

### 4.5. Migration and Wound Healing Assays

Cell migration and invasion were determined using a transwell assay. Briefly, cells were resuspended in serum-free DME/F12 and plated in the upper chamber of transwell inserts coated without BD Matrigel, and the lower chambers were filled with DME/F12 containing FBS (10%). The migrated cells were stained, imaged, and analyzed by calculating the cell numbers from five random fields. Cells were plated and grown into a confluent monolayer in six-well plates. Scratches were then generated using a pipette tip. After wounding, the cell migration process was visualized using a microscope at 0 and 48 h.

### 4.6. Real-Time PCR 

Total RNA was extracted using Trizol reagent followed by Nanodrop concentration and purity analysis. cDNA was synthesized by M-MLV Reverse Transcriptase (Promega Corporation, Beijing, China (M1708)). RT-qPCR was conducted using a SYBR green kit (Yeasen Biotechnology, Shanghai, China (11201ES03)) according to the manufacturer’s instructions. Each sample was tested in triplicate. GAPDH was used for normalization. The data were evaluated by the 2^−ΔΔCt^ method. RT-qPCR primers used are as described in Table 2.

### 4.7. Western Blotting

Proteins were obtained from cell lysates in lysis buffer and were used for protein quantification. Proteins were separated by sodium dodecyl sulfate-polyacrylamide gel electrophoresis, transferred to polyvinylidene difluoride membranes, and subjected to immunoprobing with specific antibodies. A chemiluminescence reagent kit purchased from Thermo Fisher Scientific company was used to detect the proteins. Images were obtained by the imager station (Gene Company Limited, Shanghai, China).

### 4.8. Isolation of Conditioned Medium (CM)

The medium was replaced 6 h after transfections and cells were incubated for 72 h. CM was collected, centrifuged, and immediately used for recipient cells incubation (72 h) or stored at −20 °C for later use.

### 4.9. Co-Immunoprecipitation (Co-IP) and Immunoprecipitation (IP) Analysis

Co-IP analysis was made as described [4] on 293T cells. The following antibodies were used: anti-ATF4 (Proteintech Group, Wuhan, China (10835-1-AP)), anti-Flag (Sigma, Shanghai, China (F1804)), and anti-HA (Sigma, Shanghai, China (H9658)). For Flag–TRIB3 and HA–ATF4 co-immunoprecipitation, we transfected the above two plasmids into 293T cells at the same time. After transfecting 48 h, cells were lysed with the Co-IP lysis buffer [150 mmol/L NaCl, 25 mmol/L Tris-HCI (pH 7.4), 2.5 mmol/L MgCl_2_, 0.5% NP-40, 0.5 mmol/L EDTA, 5% glycerol], the lysate was incubated with the specific antibodies at 4 °C overnight, and then with Protein A/G Plus-Agarose at 4 °C for 4 h. Interaction complexes were detached from the beads by boiling for 70 °C, 15 min, followed by SDS–PAGE and immunoblot analysis. We also did the IP analysis in MRC5 cells. We only transfected the Flag-TRIB3 plasmid in the MRC5 cells, and collect the cells also after transfecting 48 h, then repeat the above steps. TRIB3–ATF4 interaction in MRC5 cells was proved by silver staining. For silver staining, samples were run on SDS–PAGE gel and stained with a SilverQuest Silver Staining kit (thermo, Shanghai, China (WL 338261)) according to the manufacturer’s instructions.

### 4.10. Luciferase Reporter Analysis

Whether there was a direct target between the ATF4′ promoter and TRIB3 was confirmed by luciferase reporter gene assay. HEK293T cells were seeded in 24-well plates 24 h before transfection. Sequences of ATF4-TSSs (−1000 bp to +100 bp and −100 bp to +1000 bp) were inserted into the pGL3.0 luciferase reporter vector. Details for primers are provided in Table 2. To assess ATF4 transcription, TK plasmid, Flag-TRIB3 plasmid, and pGL3.0-ATF4 luciferase reporter plasmids were transfected into HEK293T cells. Luciferase activity was measured at 48 h post transfection with a Dual-Luciferase assay system as instruction described. The Dual-Luciferase Reporter kit (Yeasen Biotechnology, Shanghai, China (11402ES80)) was used to detect luciferase activity. 

### 4.11. Collagen Gel Contraction (CGC) Assay

In order to detect the proliferation ability and contractility of fibroblasts, we designed the collagen gel contraction (CGC) assay. We transfected the Flag-TRIB3 (Figure 4C) or HA-ATF4 (Figure 6C) in the MRC5 cells. Then, 48 h after plasmid transfection, we performed the CGC assay to test the lung fibroblast activation. Transfected MRC5 cells were suspended in a serum-free medium and blended with 3 mg/mL neutralized rat tail type I collagen (ratio 2:1) after transfecting. Subsequently, the mixture was seeded in 24-well plates at 1 × 10^5^/mL cell density. After collagen coagulating for 1 h at 37 °C, the gel edge was detached from well walls, and 1 mL 10% FBS containing culture medium was added to the gels. The incubation time depends on cell viability. Finally, the gel images were captured and then analyzed by ImageJ software. The gel area reflects the cell proliferation ability and contractility. The smaller the area of gel, the greater the cell contractility.

### 4.12. Gene Expression Data

Differentially expressed genes were determined by comparing IPF group to control groups using the “limma” package. The process of calculating Spearman correlation coefficient is: the variables X and Y are sorted from smallest to largest respectively, represented by the rank RX and RY; when the rank is the same due to value equality, the average rank is taken as the rank of each variable. To establish a test hypothesis, determine inspection level H0: ρ = 0; there is a correlation relationship between genes. If H1: ρ ≠ 0, there is no correlation between genes based on Benjamini and Hochberg control false positive rate. After correcting, the *p* value less than 0.1, H0 was rejected and H1 was accepted, and a correlation between the genes was constructed [36].

### 4.13. Statistical Analysis

The statistical methods used in the manuscript are summarized as follows. Statistical analysis of microarray (GSE32537) was performed by R package “limma”. Spearson rank analysis was used to analyze the correlation between *TRIB3* and *FN1*, *ACTA2*, *COL1A1*, and *COL1A2* expression. Multiple comparisons were addressed using the BH method. All experiments were carried out at least three times. The results in the control and experimental groups were analyzed by GraphPad software 9.0. The Shapiro–Wilk normality test was used to test normal distribution. The results were analyzed by the Mann–Whitney U test for comparisons between two groups when sample data were not normally distributed and by unpaired Student’s *t*-test for comparisons between two groups with normal distribution. Data are presented as mean ± SD and were considered statistically significant at *p* < 0.05. In particular, in Figure 3A, Figure 5E, and Figure 7C, significant differences between groups were evaluated by two-way ANOVA with Šídák’s multiple comparisons test for pairwise comparisons. 

## Figures and Tables

**Figure 1 ijms-23-15705-f001:**
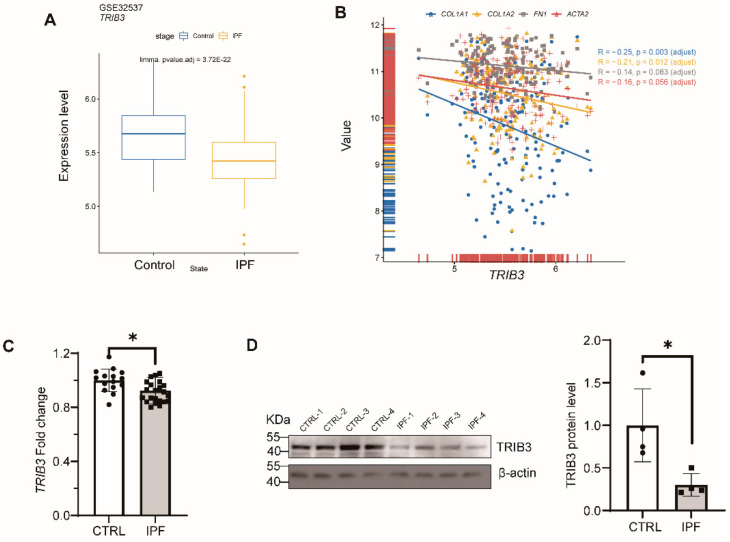
Down-regulation of *TRIB3* in IPF lung. (**A**) *TRIB3* was down-regulated in IPF patients, which was profoundly validated from the Gene Expression Omnibus (GEO) RNA-seq dataset (GSE32537). Statistical analysis of the microarray was performed by R package “limma”. (**B**) *TRIB3* was negatively correlated with *COL1A1*, *COL1A2*, *FN1*, and *ACTA2*. Data were obtained from GSE32537 (Control = 50, IPF = 119) in the public database Gene Expression Omnibus (GEO). Spearson rank analysis was used to analyze the correlation between *TRIB3* and *FN1*, *ACTA2*, *COL1A1*, and *COL1A2* expression. (**C**) *TRIB3* was down-regulated from patients with IPF in transcription level. The statistical test used was the Mann–Whitney U test for comparisons between two groups. * *p* < 0.05. (**D**) Western blot analysis showed reduced TRIB3 expression in IPF lungs (N = 4) compared with healthy lungs (N = 4). The statistical test used was the Mann–Whitney U test for comparisons between two groups. * *p* < 0.05.

**Figure 2 ijms-23-15705-f002:**
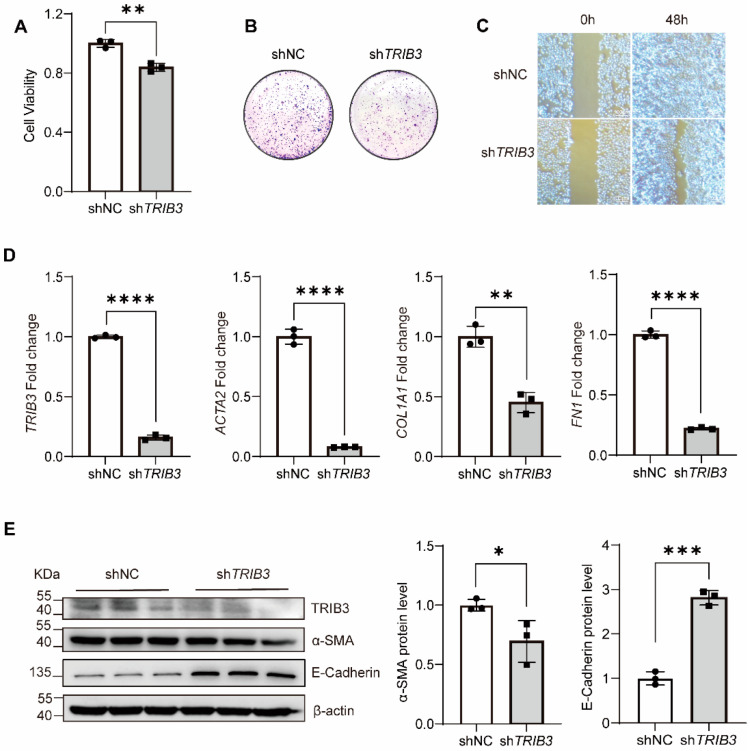
*TRIB3* deficiency inhibited A549 cell proliferation, migration, and profibrotic gene expressions. (**A**) The effect of *TRIB3* silencing on the viability of A549 cells was assessed by CCK8 assay. The results were analyzed by the unpaired Student’s *t*-test for comparisons between two groups with normal distribution. Data are presented as mean ± SD. ** *p* < 0.01. (**B**) The effect of *TRIB3* silencing on the colony formation of A549 cells (N = 3 independent experiments). (**C**) *TRIB3*-shRNA inhibited the migration of A549 cells. (**D**) Gene expression analysis by quantitative PCR (qPCR) demonstrates a significant decrease in *ACTA2*, *COL1A1*, and *FN1* transcript levels in *TRIB3*-silenced normal epithelial cells. *GAPDH* was used as an internal control. The results were analyzed by the unpaired Student’s *t*-test for comparisons between two groups with normal distribution. Data are presented as mean ± SD. ** *p* < 0.01, **** *p* < 0.0001. (**E**) *TRIB3*-shRNA decreased the protein level of TRIB3, α-SMA, E-Cadherin, and β-actin (N = 3 independent experiments). β-actin was used as an internal control. The results were analyzed by the unpaired Student’s *t*-test for comparisons between two groups with normal distribution. Data are presented as mean ± SD. * *p* < 0.05 and *** *p* < 0.001.

**Figure 3 ijms-23-15705-f003:**
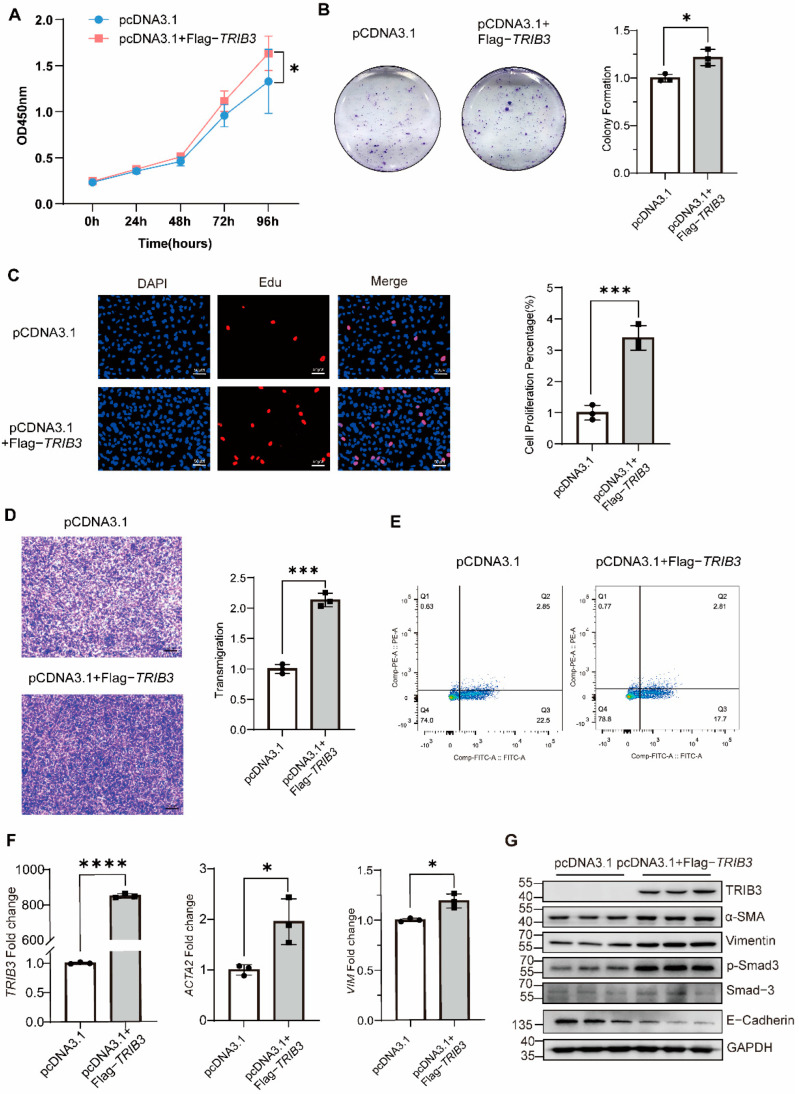
*TRIB3* expression promoted A549 cell proliferation, migration, and profibrotic genes expression. (**A**) The effect of *TRIB3* overexpression on the viability of A549 cells was assessed by CCK8 assay (N = 4 independent experiments). Significant differences between groups were evaluated by two-way ANOVA with Šídák’s multiple comparisons test for pairwise comparisons. * *p* < 0.05. (**B**) The effect of *TRIB3* overexpression on the colony formation of A549 cells (N = 3 independent experiments). The results were analyzed by the unpaired Student’s *t*-test for comparisons between two groups with normal distribution. Data are presented as mean ± SD. * *p* < 0.05. (**C**) The EdU proliferation assay was used to measure A549 cell proliferation. The results were analyzed by the unpaired Student’s *t*-test for comparisons between two groups with normal distribution. Data are presented as mean ± SD. *** *p* < 0.001. (**D**) Crystal violet staining of A549 cell is shown for the transwell assay (N = 3 independent experiments). The results were analyzed by the unpaired Student’s *t*-test for comparisons between two groups with normal distribution. Data are presented as mean ± SD. *** *p* < 0.001. (**E**) Cell apoptosis measured by cytometry, with quantification analysis (N = 3 independent experiments). (**F**) Gene expression analysis by qPCR demonstrated a significant increase in *TRIB3*, *ACTA2*, and *VIM* transcript levels in *TRIB3*-overexpressed normal epithelial cells. *GAPDH* was used as an internal control. The results were analyzed by the unpaired Student’s *t*-test for comparisons between two groups with normal distribution. Data are presented as mean ± SD. * *p* < 0.05 and **** *p* < 0.0001 vs. control. (**G**) Western blot analysis showed that the expression of α-SMA, vimentin, E-Cadherin, and p-Smad3 in A549 cells increased after overexpressing *TRIB3* (N = 3 independent experiments). For western blot, GAPDH was used as an internal control.

**Figure 4 ijms-23-15705-f004:**
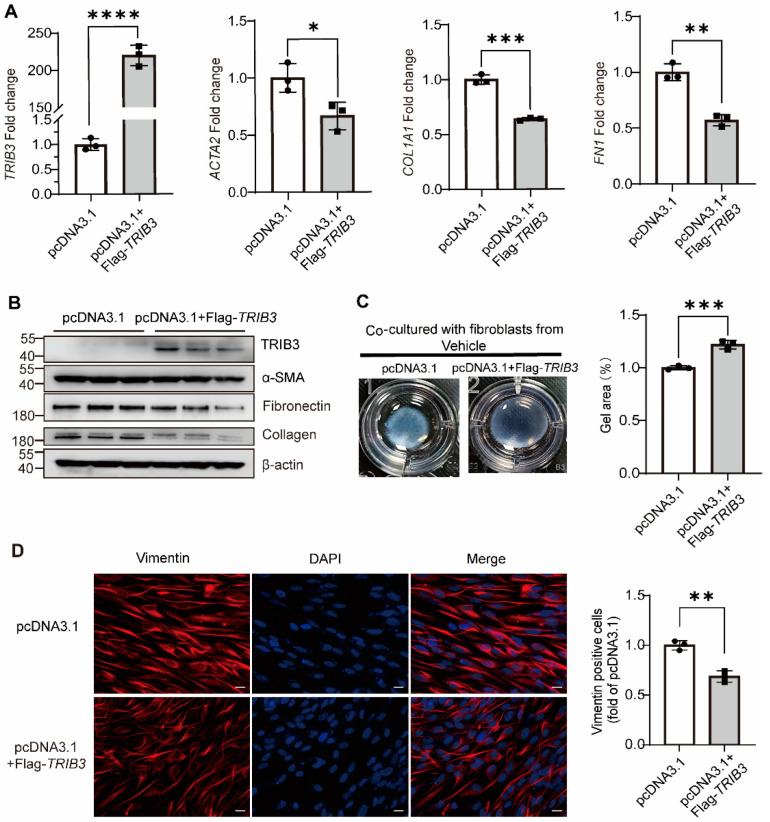
*TRIB3* inhibits lung fibroblast activation. (**A**) Gene expression analysis by qPCR demonstrates a significant decrease in *ACTA2*, *COL1A1*, and *FN1* transcript levels in *TRIB3*-overexpressing normal lung fibroblasts. *GAPDH* was used as an internal control. The results were analyzed by the unpaired Student’s *t*-test for comparisons between two groups with normal distribution. Data are presented as mean ± SD. * *p* < 0.05, ** *p* < 0.01, *** *p* < 0.001, and **** *p* < 0.0001. (**B**) Western blot analysis showed αSMA, collagen I, and fibronectin elevation in *TRIB3*-overexpressed lung fibroblasts (N = 3 independent experiments). β-actin was used as an internal control. (**C**) The activation of fibroblasts was examined based on fibroblast contraction in 3D collagen matrices (N = 3 independent experiments). We used ImageJ software to measure the area of gels. The results were analyzed by the unpaired Student’s *t*-test for comparisons between two groups with normal distribution. Data are presented as mean ± SD. *** *p* < 0.001. (**D**) MRC5 cells were subjected to immunofluorescence with a vimentin antibody. Scale bars, 20 μm. ImageJ software was used to quantify the fluorescence intensity. The results were analyzed by the unpaired Student’s *t*-test for comparisons between two groups with normal distribution. Data are presented as mean ± SD. ** *p* < 0.01.

**Figure 5 ijms-23-15705-f005:**
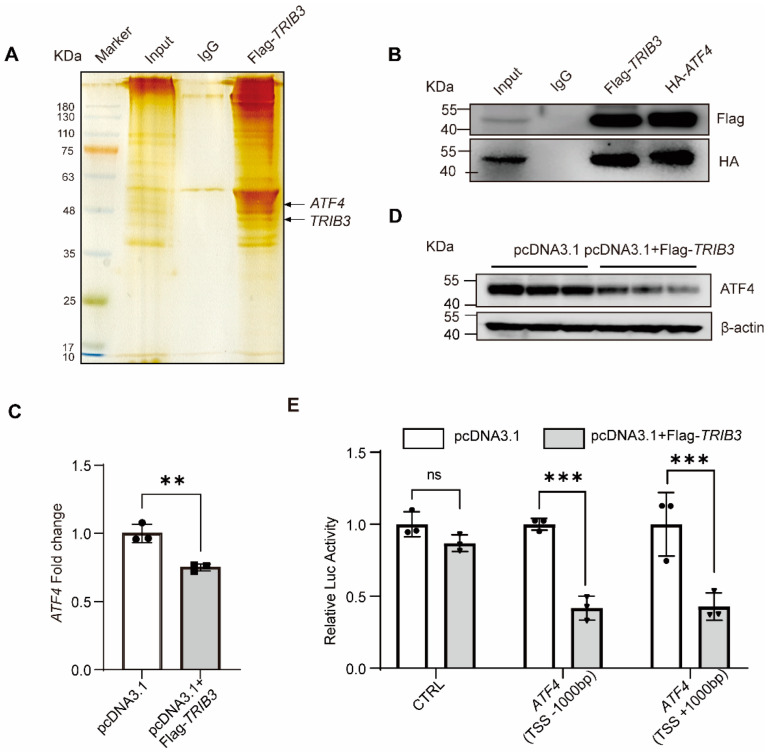
*TRIB3* interacts with *ATF4* and negatively regulates *ATF4*. (**A**) Immunoprecipitation for *TRIB3* in MRC5 cells. *TRIB3* interacts with *ATF4* by silver staining. (**B**) Co-IP of flag-tagged *TRIB3* and HA-tagged ATF4 protein from the transfected cell lysates. (**C**,**D**) Overexpression of TRIB3 decreased the expression of *ATF4* both at RNA and protein levels (N = 3 independent experiments). *GAPDH* was used as an internal control for qPCR, and β-actin was used as an internal control for western blot. The results were analyzed by the unpaired Student’s *t*-test for comparisons between two groups with normal distribution. Data are presented as mean ± SD. ** *p* < 0.01. (**E**) Relative expression of *ATF4* promoter-driven luciferase reporters in *TRIB3*-overexpressing cells (N = 3 independent experiments). Results were normalized to *Renilla* luciferase activity and expressed as relative luciferase units. Significant differences between groups were evaluated by two-way ANOVA with Šídák’s multiple comparisons test for pairwise comparisons. *** *p* < 0.001. ns, no significance.

**Figure 6 ijms-23-15705-f006:**
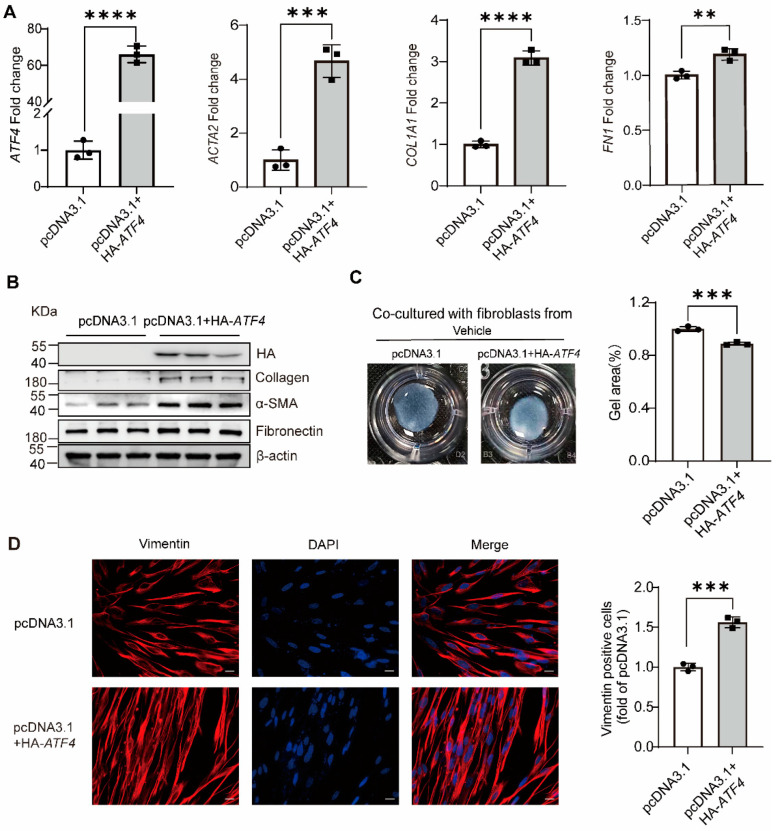
*ATF4* promotes lung fibroblast activation. (**A**) qPCR analysis of *ATF4*-overexpressing normal fibroblasts (pcDNA3.1+HA-*ATF4*) shows enhanced profibrotic transcript levels compared with normal lung fibroblasts transfected with an empty vector plasmid. The results were analyzed by the unpaired Student’s *t*-test for comparisons between two groups with normal distribution. Data are presented as mean ± SD. ** *p* < 0.01, *** *p* < 0.001, and **** *p* < 0.0001. (**B**) HA-tagged *ATF4*, collagen I, α-SMA, fibronectin, and β-actin proteins were detected by western blot (N = 3 independent experiments). (**C**) The activation of fibroblasts was examined based on fibroblast contraction in 3D collagen matrices (N = 3 independent experiments). We used ImageJ software to measure the area of gels. The results were analyzed by the unpaired Student’s *t*-test for comparisons between two groups with normal distribution. Data are presented as mean ± SD. *** *p* < 0.001. (**D**) MRC5 cells were subjected to immunofluorescence with a vimentin antibody. Scale bars, 20 μm. ImageJ software was used to quantify the fluorescence intensity. The results were analyzed by the unpaired Student’s *t*-test for comparisons between two groups with normal distribution. Data are presented as mean ± SD. *** *p* < 0.001.

**Figure 7 ijms-23-15705-f007:**
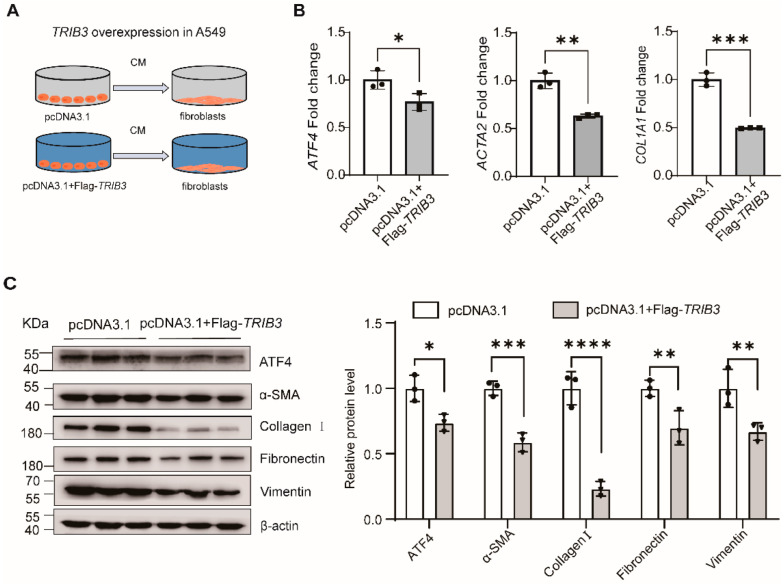
*TRIB3* attenuates lung fibrosis by negative regulation of *ATF4*. (**A**) Schematic of transfection of A549 cells before CM collection and transfer. CM from pcDNA3.1 or pcDNA3.1+Flag-*TRIB3*-transfected cells was collected 72 h after transfection and transferred to human lung fibroblast (MRC5) for an additional 72 h. (**B**) qPCR analysis showing that CM from pcDNA3.1+ Flag-*TRIB3*-transfected cells reduced the mRNA level of *ATF4* and profibrotic genes in MRC5 cells. The results were analyzed by the unpaired Student’s *t*-test for comparisons between two groups with normal distribution. Data are presented as mean ± SD. * *p* < 0.05, ** *p* < 0.01 and *** *p* < 0.001. (**C**) Western blot analysis showing CM from pcDNA3.1+ Flag-*TRIB3*-transfected cells reduced the expression of ATF4, α-SMA, Collagen I, Fibronectin, and vimentin. Significant differences between groups were evaluated by two-way ANOVA with Šídák’s multiple comparisons test for pairwise comparisons. * *p* < 0.05, *** *p* < 0.001 and **** *p* < 0.0001.

**Figure 8 ijms-23-15705-f008:**
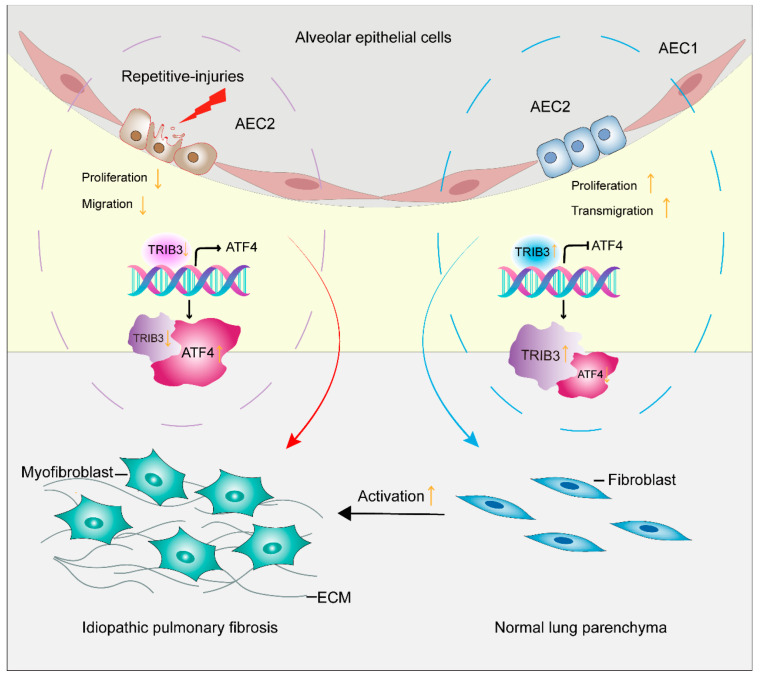
Antifibrotic mechanism of TRIB3 mediation of pulmonary fibrosis through inhibition of lung fibroblast activation. TRIB3 inhibits the transformation of pulmonary fibroblasts into myofibroblasts through interaction with ATF4 and inhibiting its expression.

**Table 1 ijms-23-15705-t001:** Primers used in the study.

Gene	Primer	Sequence 5′-3′
Flag-TRIB3	Sense	CGCGGATCCGCCACCATGGATTACAAGGATGA
		CGACGATAAGCGAGCCACCCCTCTGGCTGCT
	Anti-sense	CCGCTCGAGCTAGCCATACAGAACCACTTCTC
		TGTCTCCCTCCTCTTCCCTG
HA-ATF4	Sense	CGCGGATCCGCCACCATGACCCATACGACGTC
		CCAGACTACGCTACCGAAATGAGCTTCCTGAG
	Anti-sense	CCGCTCGAGCTAGGGGACCCTTTTCTTCCCCC
		TTGCCTTGCGGACCTCTTCT
shTRIB3	Sense	CCGGGATCTCAAGCTGTGTCGCTTTCTCGAGA
		AAGCGACACAGCTTGAGATCTTTTTG
	Anti-sense	AATTCAAAAAGATCTCAAGCTGTGTCGCTTTCT
		CGAG AAAGCGACACAGCTTGAGATC
ATF4 (TSS-1000 bp to +100 bp)	Anti-sense	CGGGGTACCTTCTGTGGCAGCCTTGCACTTGAG
		CCGGATGAAAATTGTAAAAACCC
	Sense	CCCAAGCTTCCCCTAATACGCCATGGTGGCCG
		TGGACCCTGAGGGCGGGGAGGAGG
ATF4 (TSS-100 bp to +1000 bp)	Sense	CGGGGTACCCGGGAGGAGACGGTCACGTGGT
		CGCGGCGGAAGGATGCGTCTGTGCT
	Anti-sense	CCCAAGCTTCCTCCAGGTAATCATCTAAGAG
		ACCTAGGCTTTCTTCAGCCCCCAAA

**Table 2 ijms-23-15705-t002:** Primers used in the study (qPCR).

**Gene**	**Primer**	**Sequence 5′-3′**
**TRIB3**	Sense	GCGGTTGGAGTTGGATGA
	Anti-sense	GCCACAGCAGTTGCACGA
**ATF4**	Sense	TCAAACCTCATGGGTTCTCC
	Anti-sense	GTGTCATCCAACGTGGTCAG
**ACTA2**	Sense	CTCTGGACGCACAACTGGCATC
	Anti-sense	CACGCTCAGCAGTAGTAACGAAGG
**COL1A1**	Sense	GAGGGCCAAGACGAAGACATC
	Anti-sense	CAGATCACGTCATCGCACAAC
**FN1**	Sense	ACAACACCGAGGTGACTGAGAC
	Anti-sense	GGACACAACGATGCTTCCTGAG
**VIM**	Sense	TTGCCGTTGAAGCTGCTAACTACC
	Anti-sense	AATCCTGCTCTCCTCGCCTTCC
**ACTB**	Sense	CACCATTGGCAATGAGCGGTTC
	Anti-sense	AGGTCTTTGCGGATGTCCACGT
**GAPDH**	Sense	GTCTCCTCTGACTTCAACAGCG
	Anti-sense	ACCACCCTGTTGCTGTAGCCAA

## Data Availability

All data are contained within the manuscript.

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
