# Peer review of "TRIB3 Mediates Fibroblast Activation and Fibrosis though Interaction with ATF4 in IPF"

_ijms, 2022, doi:10.3390/ijms232415705_

Round 1

Reviewer 1 Report

The authors presented a study showing the role of the TRIB3-ATF4 axis in IPF. In general, the role of both TRIB3 and ATF4 in IPF is known, but the authors have considered various aspects of TRIB3-ATF4 interactions that may be of interest to specialists working in this field.

 I have some comments as below.

Major

1.     First of all, the manuscript should be thoroughly revised regarding the style of writing. Huge compound and complex sentences are inconvenient for perception. They should be broken down into separate sentences. The manuscript is full of grammatical and stylistic errors.

2.     It is unclear in the abstract where the study justification ends and where the description of the results begins.

3.     The basic structure used by the authors makes reading challenging. For example, the last paragraph of the introduction contains the literature data, then own results, then the hypothesis being tested.

4.     Gene symbols should be written in italics.

5.     Please, give a table on patients (age, gender, stages, smoking status, disease duration, etc.). This table can be presented in supplementary materials.

6.     The Discussion should start with a brief summary of the main findings (Albert T. Winning the Publications Game: The Smart Way to Write Your Paper and Get It Published. 4th ed. CRC Press; 2016). In the present study, the discussion starts with literature data.

7.     Statistics: overall, statistical processing is weak. Statistical data should be presented more accurately and carefully. What are the hypotheses for using any statistical tests? The questions are not formulated. Such a mandatory procedure as correction for multiple comparisons has not been performed. The authors used the Pearson coefficient (Fig. 1C), but was the distribution evaluated as normal?

Minor

8.     A weighted analysis of the gene coexpression network was performed on the GSE32537 dataset. Where are these data presented? Is it Fig. 1C?

Reviewer 2 Report

The manuscript from Wang et al investigated the role of TRIB3 in fibroblast activation causing pulmonary fibrosis via its interaction with ATF4. The authors demonstrated that TRIB3 is downregulated IPF lungs and TRIB3 overexpression in A549 cells increased cell proliferation and migration. TRIB3 overexpression inhibited fibroblast activation and decreased ECM synthesis via downregulating ATF4 in MRC5 cells suggesting TRIB3 is antifibrotic. Other studies on IPF by  Liu et al, (Acta Pharmaceutica Sinica, 2021), skin fibrosis by Tomcik et al, (Ann Rheum Dis, 2016), kidney fibrosis by Wang et al (Diabetes Res Clin Pract, 2014) and liver fibrosis by Zhang et al on (Autophagy, 2020) showed profibrotic effect of TRIB3. Previous study on lung fibrosis further used a peptide to disrupt TRIB3‒GSK-3β interaction and concluded that as a novel treatment strategy for PF by targeting the TRIB3‒GSK-3β interaction in two distinct mouse models of lung fibrosis (Acta Pharmaceutica Sinica, 2021). In this earlier study, the authors focused on TRIB3 function on alveolar macrophages and stated that TRIB3 expression was elevated in AM of IPF patients and in bleo-induced mouse model of PF targeting the complex TRIB3‒GSK-3β regulatory axis could be therapeutic. All these previous studies showed TRIB3 as profibrotic and disagree with this present finding where the authors showed TRIB3 as antifibrotic. Although in this MS the authors proposed that TRIB3 may function differently in epithelial cell and fibroblast, however the fibroblasts are the ultimate cell type responsible for generating ECM causing fibrosis.

Comments

In Fig. 1D, Authors showed in Western blot analysis reduced TRIB3 expression in 4 healthy and 4 IPF lungs. Although, in the legend they claimed IPF lungs (N=24) compared with healthy lungs (N=15). A quantitative result is needed combining all the samples.

In Fig. 2E, the decrease in ASMA protein levels is not obvious. Some quantitative analysis is required.

In Fig. 7C, inhibition of ASMA and ATF4 by TRIB3 OE at the protein levels are not obvious.

Round 2

Reviewer 1 Report

In the corrected version of the manuscript, the authors partially took into account my comments, but there are still errors and inaccuracies. Some questions remained unanswered.

1.     Text editing is still required. Some, but not all, examples (p. 2, line 75) “The expression of TRIB3 were negatively correlated with disease severity in IPF patients”, (p. 2, line 89) “This significant downregulation of TRIB3 in IPF patients was profound validated from Gene Expression Omnibus (GEO) RNA-seq dataset (GSE32537) (Figure 1B).”

2.     The names of the genes in the Figures and their Legends are still not written in italics.

3.     For the most part, questions about statistics remained unanswered. Corrections for the statistical analysis are indicated only for data from GEO dataset (GSE32537). In other sections, statistical analysis should also be performed taking into account standard requirements, such as the formulation of a statistical hypothesis, the verification of distribution and the choice of method, correction for the multiplicity of comparison for all analyzed indicators in the work (that is, all tests performed on the same samples); data “*p < 0.05, **p<0.01, ***p<0.001 and ****p<0.0001 vs control” should be given after correction for multiple comparisons.

Author Response

Response to Reviewer 1 Comments

Comments and Suggestions for Authors

In the corrected version of the manuscript, the authors partially took into account my comments, but there are still errors and inaccuracies. Some questions remained unanswered.

Point 1: Text editing is still required. Some, but not all, examples (p. 2, line 75) “The expression of TRIB3 were negatively correlated with disease severity in IPF patients”, (p. 2, line 89) “This significant downregulation of TRIB3 in IPF patients was profound validated from Gene Expression Omnibus (GEO) RNA-seq dataset (GSE32537) (Figure 1B).”

Response: Done, we appreciate your comment and rephrased this part as the revised manuscript.

Point 2: The names of the genes in the Figures and their Legends are still not written in italics.

Response: Done, we changed the names of the genes in the Figures and their Legends. See the revised manuscript.

Point 3: For the most part, questions about statistics remained unanswered. Corrections for the statistical analysis are indicated only for data from GEO dataset (GSE32537). In other sections, statistical analysis should also be performed taking into account standard requirements, such as the formulation of a statistical hypothesis, the verification of distribution and the choice of method, correction for the multiplicity of comparison for all analyzed indicators in the work (that is, all tests performed on the same samples); data “*p < 0.05, **p<0.01, ***p<0.001 and ****p<0.0001 vs control” should be given after correction for multiple comparisons.

Response: We thank the reviewer for the comments. The statistical methods used in the manuscript are summarized as follows. Statistical analysis of microarray (GSE32537) was performed by R package “limma”. Spearson rank analysis was used to analyze the correlation between TRIB3 and FN1, ACTA2, COL1A1, and COL1A2 expression. Multiple comparisons were addressed using the BH method. All experiments were carried out at least three times. The results in the control and experimental groups were analyzed by GraphPad software 9.0. The Shapiro-Wilk normality test was used to test normal distribution. The results were analyzed by the Mann-Whitney U test for comparisons between two groups when sample data were not normally distributed, and by unpaired Student’s t-test for comparisons between two groups with normal distribution. Data are presented as mean ± SD and were considered statistically significant at P < 0.05. In particular, in Fig. 3A, Fig. 5E, and Fig. 7C, significant differences between groups were evaluated by two-way ANOVA with Šídák's multiple comparisons test for pairwise comparisons.

Reviewer 2 Report

Although the authors responded to my comments however they didn't provide any explanations regarding my comments on previous findings which contradicts this current finding

"previous studies showed TRIB3 as profibrotic and disagree with this present finding where the authors showed TRIB3 as antifibrotic."

Author Response

Response to Reviewer 2 Comments

Comments and Suggestions for Authors

Although the authors responded to my comments however they didn't provide any explanations regarding my comments on previous findings which contradicts this current finding.

"previous studies showed TRIB3 as profibrotic and disagree with this present finding where the authors showed TRIB3 as antifibrotic."

Response: Thanks for your comments. Pulmonary fibrosis is also complex and appreciated as a disease of change—a condition characterized by plasticity and heterogeneity, that evolves at genetic, phenotypic and pathological levels, and progresses through different stages clinically. Beyond decoding of the genetic fingerprint and molecular makeup of different cells, we understand the importance of the systemic and local fibrotic microenvironment in how the disease develops and manifests. In condition of our experiments, we showed TRIB3 was antifibrotic mediator.